# Opportunities for Improving Waterlogging Tolerance in Cereal Crops—Physiological Traits and Genetic Mechanisms

**DOI:** 10.3390/plants10081560

**Published:** 2021-07-29

**Authors:** Cen Tong, Camilla Beate Hill, Gaofeng Zhou, Xiao-Qi Zhang, Yong Jia, Chengdao Li

**Affiliations:** 1Western Crop Genetic Alliance, College of Science, Health, Engineering and Education, Murdoch University, 90 South Street, Murdoch, WA 6150, Australia; 33972313@student.murdoch.edu.au (C.T.); Camilla.Hill@murdoch.edu.au (C.B.H.); turfs@163.com (G.Z.); Xiao-Qi.Zhang@murdoch.edu.au (X.-Q.Z.); Y.Jia@murdoch.edu.au (Y.J.); 2Western Australian State Agricultural Biotechnology Centre, Murdoch University, 90 South Street, Murdoch, WA 6150, Australia; 3Department of Primary Industries and Regional Development, 3-Baron-Hay Court, South Perth, WA 6151, Australia

**Keywords:** waterlogging tolerance mechanism, *Arabidopsis*, rice, maize, wheat, barley, QTL, candidate genes

## Abstract

Waterlogging occurs when soil is saturated with water, leading to anaerobic conditions in the root zone of plants. Climate change is increasing the frequency of waterlogging events, resulting in considerable crop losses. Plants respond to waterlogging stress by adventitious root growth, aerenchyma formation, energy metabolism, and phytohormone signalling. Genotypes differ in biomass reduction, photosynthesis rate, adventitious roots development, and aerenchyma formation in response to waterlogging. We reviewed the detrimental effects of waterlogging on physiological and genetic mechanisms in four major cereal crops (rice, maize, wheat, and barley). The review covers current knowledge on waterlogging tolerance mechanism, genes, and quantitative trait loci (QTL) associated with waterlogging tolerance-related traits, the conventional and modern breeding methods used in developing waterlogging tolerant germplasm. Lastly, we describe candidate genes controlling waterlogging tolerance identified in model plants *Arabidopsis* and rice to identify homologous genes in the less waterlogging-tolerant maize, wheat, and barley.

## 1. Introduction

In the past few decades, climate change has increased the probability of extreme weather events such as drought and floods occurring [1]. It was estimated that 10–12% of the agricultural regions worldwide experience flooding of farming land, with annual losses of more than $74 USD billion [2,3]. In barley (*Hordeum vulgare* L.), reduction of yield due to waterlogging stress ranges from 20% to 25% depending on the duration of waterlogging, soil type, varieties, and stage of plant development [4]. About 15% of maize (*Zea mays* L.) production area is prone to waterlogging, leading to 20–30% yield loss [5]. About 10–15 million ha of wheat (*Triticum aestivum* L.) sown area is affected by waterlogging, which causes 20–50% of yield losses [6]. More than 16% of rice (*Oryza sativa* L.) land in the world is affected by flash-flooding and submergence [7]. The development of waterlogging tolerance varieties is important to maintain crop production in areas prone to flooding and submergence due to high rainfall and poor drainage. Understanding the detailed mechanisms of waterlogging tolerance is crucial for breeders to target different tolerance-related traits and to develop new varieties with greater waterlogging tolerance.

Two types of flooding can occur in the field: (1) waterlogging, the saturation of the soil surface with the root system under water, and (2) submergence, where the plant is partially or entirely immersed in water. For wetland plants, such as lowland rice varieties adapted to the lowland growing ecologies can withstand 5–25 cm waterlogging stress in the field but are vulnerable to complete submergence. Under deep-water flooding, most rice varieties exhibit a unique coping strategy, reserving energy by ceasing internode elongation and regrowth after flood waters recede [4]. In barley, wheat, and maize, waterlogging stress causes energy shortage, disturbs root hydraulic conductance, reduces nutrient uptake, and decreases photosynthesis, leading to significant yield losses [8]. Plants adapt to waterlogging stress by enhanced anaerobic respiration due to lower oxygen diffusion in water. Higher enzymatic activity of ethanol fermentation and involvement of antioxidant defence mechanisms are adaptive traits for plants to produce energy and cope with post-hypoxia oxidative stress under waterlogged conditions [9,10]. Phytohormones signalling (ethylene, abscisic acid, and gibberellin) regulates aerenchyma formation and internode elongation [11].

Previous review papers primarily summarised waterlogging tolerance mechanisms, genes, and quantitative trait loci (QTL) associated with tolerance-related traits. In this review, we reviewed (1) the current knowledge on waterlogging tolerant mechanisms in the four major cereal crops (rice, maize, wheat, and barley); (2) the methods used for screening germplasm for waterlogging tolerance, (3) identified the syntenies between candidate genes detected in *Arabidopsis*, rice, maize, wheat, and barley to explore the potential of some genes and major effects QTL in developing waterlogging tolerance varieties using marker-assisted selection, and (4) highlighted some of the challenges associated with QTL mapping.

## 2. Physiological Mechanisms of Plant Responses to Waterlogging Stress

### 2.1. Oxygen Deprivation

Excessive water causes oxygen shortage, which adversely affects root growth, shoot growth, photosynthesis, hydraulic conductivity as well as nutrient uptake. Oxygen deprivation is the main constraint under waterlogging stress as gas diffusion is 10^4^ times slower in water than in air [12]. Lower oxygen availability decreases plant respiration rate and adenosine triphosphate (ATP) production which reduces root growth [13]. Plant wilting is caused by decreased respiration and ATP synthesis loss in waterlogged roots [10]. Glucose is the primary fuel for glycolysis and downstream pathways such as respiration to produce energy for plant growth and reproduction [14]. During the respiration process, glucose enters the glycolysis pathway to produce pyruvate and two ATP molecules. Then pyruvate is combusted to CO_2_ and H_2_O in the mitochondria to produce high energy (36 ATP) with sufficient oxygen supply as part of the tricarboxylic acid (TCA) cycle. Under hypoxic conditions, pyruvate is fermented to ethanol on the cytoplasm, releasing two molecules of ATP [15].

The inhibition of root growth due to energy deficiency was observed in waterlogged wheat, barley, maize, and rice. A study in wheat and barley found that root and shoot growth were significantly inhibited after 11 days of waterlogging treatment [16]. The dry weight of waterlogged shoots and roots as well as the ratio of root/shoot were significantly lower than the well-drained plants [17]. In barley, waterlogging stress caused a marked decrease in shoot and root dry weight with a more severe effect in susceptible varieties than the tolerant genotypes [18]. In maize, waterlogging restricted root development and accelerated root senescence, which caused a significant reduction in shoot and root dry weight [19]. Hypoxia also decreased root elongation and dry weight in both lowland and upland rice varieties [20]. The limited root growth of waterlogged plants also restricted the absorption area for water and nutrient uptake [19].

### 2.2. Photosynthesis Rate

The photosynthetic rate of the four crops is reduced by waterlogging stress due to stomatal closure, mesophyll conductance, chlorophyll degradation, damage of photosystem II (PS II), and reduced photosynthetic enzyme activity [21]. Ploschuk et al. found that the net photosynthesis rate of waterlogged barley plants firstly decreased up to 40% after 6 h of waterlogging treatment [21]. Rice is a resilient crop for waterlogging, but it also exhibited decreased photosynthetic rate by 50% after 4 days of anoxic treatment [22]. Transpiration rate and stomatal conductance were shown to decline under waterlogging stress [23]. Stomatal closure was found to be correlated with transpiration and CO_2_ exchange rate, which is responsible for the decrease in photosynthesis capacity under waterlogging conditions [24]. In wheat, the stomatal closure decreased net photosynthesis by limiting internal CO_2_ concentration [25]. The photosynthetic rate also decreased due to damaged mesophyll cell ultrastructure and reduced chlorophyll content. Chloroplast structure in mesophyll cells is the fundamental component for normal leaf photosynthesis and was found to be damaged in waterlogged maize [26]. Lower chlorophyll (a + b) content was also found in flooded maize plants, which was about 20% smaller than the control plants [27]. Waterlogged barley exhibited lower chlorophyll content than control plants, with the reduction more pronounced in waterlogging-sensitive than the tolerant varieties [28]. Photosynthesis processes, such as energy transfer, light absorption, and photochemical reactions occurred in PS II which can be determined by chlorophyll fluorescence parameters [29]. Waterlogging destructs chloroplast structure and continuously inhibits photosynthetic electron transport as well as PS II activity [27]. With a longer waterlogging duration, photosynthetic enzyme activity is further reduced. A noticeable reduction in net photosynthesis and ribulose-1,5-bisphosphate carboxylase (RuBisCo) activities in barley plants occurred after five days of waterlogging treatment [30]. In maize plants, RuBisCo activity declined by 20–30% during waterlogging treatment [27].

### 2.3. Root Hydraulic Conductance

Wilting is a common response to waterlogging stress, which is caused by decreased root water uptake and impaired root hydraulic conductance (L_p_) [25]. L_p_, the per-unit driving force of water flow rate into roots, is correlated with transpiration rate and determines water uptake capacity [31]. Root cell death reduces L_p_ by creating physical barriers for water flow under long-term waterlogged conditions [32]. Additional factors for a significant change of L_p_ are anaerobic respiration due to lack of oxygen (see section “Energy metabolism”) as well as aquaporin gating [33]. Aquaporin is an intrinsic membrane protein, which facilitates water uptake through membrane proteinaceous pore formation, and is regulated by energy production and cytosolic pH [34]. The inhibition of aquaporin gating of waterlogged plant roots is regulated by cellular acidosis and phosphorylation of aquaporins which is caused by CO_2_ accumulation from respiration and depletion of ATP [34,35].

Waterlogging and low ambient oxygen reduces L_p_ in plants, but responses vary between species depending on the water transport pathway [32]. Three pathways of water transportation exist: (1) apoplastic (around protoplasts), (2) symplastic (through plasmodesmata), and (3) transmembrane (across membranes). The apoplastic pathway depends on root anatomy and cell wall properties, while the transmembrane pathway is controlled by aquaporins [36]. Lower L_p_ was found in *Arabidopsis*, wheat, maize under hypoxic conditions as cellular acidosis inhibits aquaporin activities [31]. However, the major pathway in some species is apoplastic, therefore the reduction in aquaporin activities under waterlogging stress has little effect on root L_p_ [32,33]. In addition, morphological changes in rice such as the formation of physical barriers to avoiding oxygen diffusion from roots may negatively influence root hydraulics [34].

### 2.4. Nutrient Absorption

A common visible sign of waterlogging stress is leaf chlorosis, which promotes early leaf senescence to remobilise nitrogen (N) to new leaves. The lower nutrient concentration of waterlogged shoots is caused by reduced nutrient uptake and transfer from roots [25]. Waterlogging decreases nutrient uptake of roots by limited surface, impaired function, reduced proton motive force, less negative membrane potential, and declined xylem loading. In particular, waterlogging stress significantly reduces N, phosphorus (P), potassium (K), magnesium (Mg), copper (Cu), zinc (Zn), and manganese (Mn) concentrations in wheat and barley [16]. The nutrient uptake by wheat seminal roots was lower in stagnant solution compared with aerated conditions [37]. In the barley root mature zone, the hypoxia immediately decreased net K^+^ uptake within a few minutes [38]. Waterlogging stress also decreased N metabolism and assimilation at different growth stages in maize [39]. Unlike waterlogging susceptible maize, barley, and wheat that showed a significant decrease in nutrient content, rice copes with waterlogging stress by forming a physical barrier to avoid oxygen diffusion from roots, which may lower the capacity of nutrient absorption by roots [40].

Three pathways exist for nutrient absorption by roots: (1) interception when roots randomly grow to explore new volumes of soil to meet nutrient requirements, (2) mass flow that describes ion transport to the root surface along with water movement driven by transpiration and evaporation, and (3) diffusion that refers to the chemical potential gradient promoting nutrient movement [41]. Inhibition of root elongation under waterlogging stress significantly decreased potential nutrient uptake by decreasing interception of nutrients [42]. Impaired root function decreases root nutrient absorption ability under waterlogged conditions. In maize, most roots (except adventitious roots) were unable to absorb nutrients from ambient soil after 6 days of waterlogging treatment [43]. Nutrient uptake by mass flow was also found to be decreased in another study, as stomatal closure reduced transpiration, water flow rate, and mass flow of nutrients [42]. Most nutrient absorption depends on diffusion and is driven by membrane potential and proton motive force, which is inhibited under waterlogging stress. The inhibited proton motive force and depolarised plasma membrane were caused by limited ATP supply, which decreased the function of the plasma membrane proton-pumping ATPase and as a consequence lowered the cytoplasmic pH [42]. Ions taken up by roots are transported to shoot through the xylem with energy supplied by plasma membrane H^+^-ATPase. Under waterlogged conditions, xylem transport is inhibited by the reduction of H^+^-ATPase in xylem parenchyma, which continuously decreases the nutrient concentration of shoot in waterlogged plants [44]. However, plants can adapt to nutrient shortages by redistributing endogenous nutrients [45]. Previous studies show that supplying extra nutrients to waterlogged barley leaves increased root N and K concentrations, indicating that nutrients are translocated from leaves to roots to compensate for losses in root uptake and support root growth under flooding conditions [46].

## 3. Adaptation and Signalling to Waterlogging Stress

### 3.1. Anatomical Adaptation

#### 3.1.1. Aerenchyma Formation

Aerenchyma, an airy tissue in adventitious roots of some plants that form intercellular spaces transporting gas between roots and shoots, is a common adaptive trait associated with waterlogging tolerance [47]. Schizogenous and lysigenous aerenchyma are two different types of aerenchyma, which are developed by cell separation and subsequent lysis of cells, respectively [48]. Lysigenous aerenchyma is formed in the root cortex of major cereal crops including rice, maize, wheat, and barley [49]. For wetland plant rice, lysigenous aerenchyma is constitutively developed under well-drained soil conditions and increases under waterlogged conditions. However, in terrestrial plants wheat, maize, and barley, the aerenchyma formation is only induced by waterlogging stress [50]. Well-developed aerenchyma was formed at 10 mm from root tips in wheat roots after 72 h waterlogging treatment [51]. In barley, aerenchyma was observed about 6 cm from the root apex of a mature root zone in the tolerant varieties after 7 days of waterlogging treatment [52]. A study of maize found cell death initially started at 10 mm from tips with aerenchyma completely formed at 30–40 mm from tips under waterlogging stress [48].

The aerenchyma formation and higher root porosity are important adaptive traits contributing to waterlogging tolerance [53]. Reactive oxygen species (ROS) and gaseous phytohormone ethylene are involved in lysigenous aerenchyma induction by initiating the programmed cell death of specific cell types. Waterlogging stress accumulates ethylene in roots due to the impeding of gas movement to the rhizosphere, as well as enhanced ethylene biosynthesis [54]. To cope with the adverse effects of ROS accumulation, antioxidant defence systems are employed in response to waterlogging stress [29].

#### 3.1.2. Adventitious Root Growth

A decreased ratio of root/shoot of waterlogged plants is due to inhibition of seminal root growth. Seminal and adventitious roots are two major types of roots in plants. Seminal roots only grow a well-developed main root axis, while adventitious roots have more central metaxylem and cortical cell layers [55]. Adventitious root development is a typical responsive trait of waterlogged plants, which can partially replace the damaged seminal root and develop more aerenchyma to improve the ability for internal oxygen transportation [44]. In greenhouse experiments, *Zea mays* ssp. *huehuetenangensis* seedlings exhibited higher adaptability to submergence with adventitious root formation [56]. The number of adventitious roots in barley tolerant genotypes was significantly greater after 21 days of waterlogging treatment than the sensitive genotypes [18]. It was shown that aerenchyma occupied 20–22% and 19% of adventitious roots in wheat and barley [21]. A study in rice found that the direction of adventitious root growth is determined by hormone auxin gradient in root tips [57]. Adventitious roots grow upward to get closer to the oxygen-rich water surface to facilitate water and nutrient uptake from the upper surface of the waterlogged soil [58,59]. In addition, adventitious root can reduce the distance of oxygen transportation between shoots and roots as it forms at the nodes of stems [60]. Adventitious roots development from the nodal epidermis is enhanced through induction of epidermal cell death driven by ethylene and ROS [61].

#### 3.1.3. Radial Oxygen Loss (ROL) Barrier

In addition to aerenchyma development, a barrier to ROL is another important responsive trait for waterlogging stress. The ROL barrier is an apoplastic barrier in the outer cell layer of roots that restricts oxygen leakage from the aerenchyma to anaerobic soil [54]. In general, the waterlogging-sensitive crops maize, wheat, and barley do not form ROL barriers, whereas rice forms ROL barriers under waterlogged or stagnant conditions [62]. The ROL barrier helps plants to maintain high oxygen levels in root tips under hypoxic soil. A study of *Zea nicaraguensis* (a wild relative of maize) found ROL barrier was developed along with the lateral and adventitious roots under hypoxic conditions [62,63]. ROL barrier development is controlled by the root length. In rice, ROL barrier formation commenced within a few hours and was well formed within 24 h in long adventitious roots (105–130 mm), while the short roots (65–90 mm) took more than 48 h for barrier formation [64]. For very short rice adventitious roots, only some of them formed ROL barrier [62].

The components of the ROL barrier that physically prevent oxygen leaks are suberin and lignin which act as diffusion barriers in the outer part of roots. ROL is controlled by the formation of suberised hypodermis and lignified sclerenchyma in roots [65]. For waterlogging-tolerant wild maize (*Zea nicaraguensis*), light ROL barrier formation was induced under deoxygenated conditions, and suberin and lignin were observed in exodermis and epidermis, respectively [66]. When rice is grown under waterlogged conditions for 2–3 weeks, both suberised and lignified cells can be observed in the basal part of roots [67]. Microarray analysis on rice adventitious roots found many putative suberin biosynthesis-related genes that were significantly upregulated during ROL barrier formation, whereas a few genes associated with lignin biosynthesis were induced [68]. Metabolite analysis of rice adventitious roots showed that malic acid and long-chain fatty acids were accumulated during ROL formation and are associated with suberin synthesis [69]. Caffeic acid o-methyltransferase (COMT), an enzyme involving in lignin biosynthesis was significantly induced in barley waterlogging tolerant varieties to form mechanical barrier via lignin deposition under waterlogging stresses [70].

### 3.2. Physiological Adaptation

Mitochondrial respiration is inhibited under waterlogging stress due to the low availability of oxygen in waterlogged soil. Oxygen deficiency switches ATP production from the TCA cycle and oxidative phosphorylation to ethanol fermentation [71]. Pyruvate from glycolysis is used for anaerobic fermentation. There are two ways for pyruvate fermentation: either producing lactic acid by lactate dehydrogenase (LDH) or acetaldehyde by pyruvate decarboxylase (PDC), which is then reduced to ethanol and regenerated nicotinamide adenine dinucleotide (NAD^+^) by alcohol dehydrogenase (ADH) [72]. ADH and PDC are important key enzymes in ethanol fermentation, which can reflect plant waterlogging tolerance [73].

ADH transcript level and activity significantly increased in maize, barley, and wheat seedlings under oxygen deficiency conditions [74,75,76]. A similar finding was found in rice, with enhanced PDC levels increasing ethanol fermentation and waterlogging survival rate [77]. However, overproduction of PDC leads to acetaldehyde accumulation, which is toxic for plants. A study on submerged rice found a negative relationship between survival rate and acetaldehyde production [78]. PDC is a rate-limiting enzyme for ethanol fermentation in flooded roots due to the toxic levels of acetaldehyde accumulation. A high ratio of ADH/PDC is required to prevent the potential acetaldehyde toxicity [76].

### 3.3. Waterlogging Stress Signalling

#### 3.3.1. Phytohormone Signalling

Ethylene is an important phytohormone that regulates plants growth and senescence and was found to accumulate under waterlogging conditions [79]. Ethylene accumulation is caused by the establishment of a barrier of plant root, which reduces the diffusion of ethylene [2]. In addition, two enzymes, namely 1-amino-cyclopropane-1-carboxylic acid (ACC) synthase and ACC oxidase, are responsible for ethylene synthesis [48,80], which were found to increase under waterlogging stress [81]. Maize roots were found to respond to environmental stress by increasing ACC synthase activity and ethylene biosynthesis [82]. Pre-treatment with ethephon, an ethylene-releasing agrochemical improved aerenchyma formation at root tips and delayed waterlogging-caused wilting in barley [83].

Ethylene regulates gas space (aerenchyma) formation in roots to help plants facilitate oxygen transportation from shoots to roots under hypoxic conditions [60]. Lysigenous aerenchyma formation is regulated by ethylene accumulation via induction of programmed cell death. The activity of enzymes involved in cell death can be changed by waterlogging stresses. Cellulase activity in germinated maize seedlings is induced by ethylene after several hours of waterlogging treatment, which contributes to cell wall decomposition to form aerenchyma [84]. The putative cell-wall-loosening enzyme xyloglucan endo-transglycosylase (XET) is also triggered by ethylene accumulation under hypoxia formation [85]. An upregulated transcript level of *XET* expression was detected in waterlogged barley and maize roots [18]. Thus, the induction of cellulase and *XET* expression is mediated by ethylene and contributes to aerenchyma formation in roots via cell wall dissolution.

In waterlogged plants, ethylene, abscisic acid (ABA), and gibberellin (GA) play an important role in survivability by mediating shoot elongation. GA promotes internode elongation by inducing the degradation of growth-inhibiting proteins [86] as well as by breaking down starch and loosing cell walls to mobilise food material to support plant growth [87]. ABA is a plant growth regulator involved in transpiration, germination, dormancy, and adaptation to stress [88,89]. GA and ABA act as antagonists in plant growth responses, with GA playing an important role in stimulating shoot growth while ABA inhibiting root elongation [81]. The application of ABA in rice decreased internode elongation induced by ethylene and GA [90]. In deep-water rice, ABA declined by 75% and GA1 increased four-fold within three hours of the waterlogged plants with ethylene treatment [91]. The adventitious roots of waterlogged wheat showed a reduction of gene expression level in ABA biosynthesis and stem node ABA content [61]. The reduction of ABA content was found in leaves and roots of both tolerant and susceptible barley varieties after three weeks of waterlogging treatment with a greater reduction in tolerant varieties [18].

#### 3.3.2. Accumulation of Reactive Oxygen Species (ROS)

Although ROS can be detrimental for plants as they unrestrictedly oxidate cell components, they also function as important secondary messengers for plant stress, including drought, salt, chilling, and mechanical stress [92,93]. In plants, ROS metabolism occurs in different organelles, including mitochondria for respiration, chloroplasts for photosynthesis, and peroxisomes for photorespiration [94]. Hydrogen peroxide (H_2_O_2_) is produced under oxygen deficiency via the disruption of the electron transport chain in mitochondria [72] and is a signal for aerenchyma formation via epidermal cell death to protect plants under anoxic stress [95,96]. In flooded rice, H_2_O_2_ application promoted lysigenous aerenchyma formation via cell death processes [97]. Similarly, H_2_O_2_ accumulation was found in wheat and barley roots under waterlogged conditions [98]. ROS accumulation is a signal for waterlogging adaptation in wheat seedlings via controlling aerenchyma formation and gene expression (*ADH* and *PDC*) involved in ethanol fermentation [99].

ROS accumulation is regulated by respiratory burst oxidase homolog (RBOH), which encodes a plasma membrane-associated nicotinamide adenine dinucleotide phosphate (NADPH) oxidase for H_2_O_2_ generation [100]. The expression of *RBOH* in crops is induced by waterlogging stress. *RBOH* expression was significantly induced in waterlogged barley roots with higher expression in tolerant varieties than in sensitive varieties [18]. In wheat roots, three *RBOH* genes were enhanced by ethylene precursor pre-treatment activated lysigenous aerenchyma formation under stagnant conditions [58]. Similarly, a study of maize found upregulation of *RBOH* gene expression leads to accumulation of ROS in cortical cells [50]. Epidermal cells of rice adventitious roots were determined by *RBOH* activity and ethylene associating with aerenchyma formation under hypoxia [101]. Glutathione S-transferase (GST) is a detoxification enzyme catalysing glutathione-dependent detoxification reaction, which was induced in waterlogged barley roots to minimise the damage of ROS accumulation [74].

A schematic diagram of the waterlogging adaptive response of energy metabolism, hormonal regulation and ROS accumulation is shown in Figure 1.

## 4. Genetic Mechanisms of Waterlogging Tolerance

Waterlogging tolerance is a complex quantitative trait, influenced by temperature, the severity and duration of stress, and plant development stage. A study of *Arabidopsis* showed that lower temperatures caused less damage under waterlogged conditions by regulating hypoxia-related genes [102]. In wheat, the severity of damage under waterlogging stress was influenced by the depth of waterlogging. The tillering ability was decreased by 62%, 45%, and 24% when the water level was at 0, 10, and 20 cm below soil surface [103]. Phenotyping of waterlogging tolerance in barley double haploid (DH) lines demonstrated a greater effect of waterlogging at the early stages [104]. QTL mapping studies conducted under waterlogging conditions have shown the effect of treatment times on QTL detection and gene expression [105] and the effect of genotype by environment (G × E) interactions. A study conducted in lowland rice using 37 genotypes across 36 environments from 1994 to 1997 showed a broad adaptation across different environmental conditions [106]. As plant phenotypes are sensitive to environmental conditions, the tolerant varieties should be evaluated in multiple conditions to identify stable waterlogging adaptive traits [107]. Due to the variation of field conditions and complexity of waterlogging tolerance, it is difficult to make direct selections in the field conditions [52].

Diverse germplasm collections are important resources for waterlogging tolerance-related traits identification [58]. There are several criteria for barley waterlogging tolerant varieties screening: plant survival rate, leaf chlorosis, and biomass reduction after waterlogging treatment [104,108]. Photosynthesis rate can also be used to characterise waterlogging tolerance as it reflects the overall performance of waterlogging tolerance level by chlorophyll content, gas exchange, and mesophyll cell ultrastructure of the youngest fully expanded leaf [18,27].

Adventitious root growth is a common responsive trait for waterlogging stress as it can switch root mass from lower ground to higher ground to reach aerated zones and form aerenchyma for oxygen transportation [109]. Aerenchyma formation, root porosity, and adventitious roots growth are important criteria for waterlogging tolerant varieties screening with higher root porosity and faster aerenchyma development [18,83]. Root porosity is defined as the percentage of gas-filled volume per unit tissue volume, which can be measured via vacuum infiltration of gas spaces [110,111]. The capacity for root aerenchyma formation can be directly observed through root cross-section or indicated by root porosity.

## 5. QTLs Associated with Waterlogging Tolerance

### 5.1. Maize

Several QTLs associated with leaf injury, adventitious root growth, and dry weight under waterlogged conditions have been reported in maize (Table 1 and Appendix A). Table 1 summarises the major effect QTL that individually explained >20% of the phenotypic variance. In F_2_ population derived from B64 × *Zea nicaraguensis*, four QTLs for aerenchyma formation under non-waterlogging conditions were mapped on chromosomes 1 (*Qaer1.02-1.03* and *Qaer1.07*), 5 (*Qaer5.09*), and 8 (*Qaer8.06-8.07*) and explained up to 46.5% of the phenotypic variation (PV) [112]. A major QTL on chromosome 4 was also mapped, which explained up to 49% PV of leaf chlorosis after 8 days of treatment in the maize population derived from *Zea nicaraguensis* and Mi29 [113]. Another QTL, *Submergence Tolerance 6* (*Subtol6*) was detected in chromosome 6 in a population of Mo18W and B73, which explained 22% of PV [114]. For adventitious root growth, three QTLs were detected on chromosomes 3 (*Qarf3.07-3.08*), 7 (*Qarf7.04-7.05*), and 8 (*Qarf8.05*) in F_2_ population derived from B64 × Na4, which together accounted for 44% of PV; *Qarf7.04-7.05* accounted for 21% of phenotypic variance [115]. In F_2_ population of B64 × *Zea mays* ssp. *huehuetenangensis*, a major QTL was detected on chromosome 8, which explained 25% PV for adventitious roots formation [56]. The QTLs for root/shoot dry and fresh weight were identified on chromosomes 5 and 9, explaining 6.3–12.0% and 30% PV under waterlogged conditions [43,116].

### 5.2. Rice

Rice has two different strategies for adapting to flooding: escape and quiescence [88]. When rice experiences deep-water flooding, internode elongation is promoted to keep the leaves above the water surface and maintain respiration and photosynthesis. However, when the plants are fully submerged, they use a quiescence strategy, ceasing shoot elongation to save energy and carbohydrates for regrowth after the water recedes [117]. The two strategies are controlled by different QTLs. Table 1 and Appendix A summarise the major QTLs associated with the waterlogging tolerance in rice. Ten QTLs were identified on chromosomes 1, 3, 8 and 12 for internode elongation regulation, which individually explained 14–36% PV in cultivated and wild rice [118]. A major QTL was detected on chromosome 12 for internode elongation under deep water conditions [119]. *Submergence-1* (*Sub1*) locus was identified on chromosome 9 using F_3_ population derived from a cross between a tolerant indica rice line (IR40931-26) and a susceptible japonica line (PI543851). IR40931-26 had the favourable allele for tolerance, which originated from an unimproved FR13A and explained up to 69% PV [120]. *Sub*1 consists of a cluster of three genes (*Sub1A*, *Sub1B*, *Sub1C*) that encodes the ethylene response factor. *Sub1A* is considered the most important gene for waterlogging tolerance as it was found to be genetically diverse among rice accessions, whereas both *Sub1B* and *Sub1C* were identical across all analysed accessions [121].

### 5.3. Wheat

Several QTLs have been identified in wheat for waterlogging tolerance traits. The major QTLs are summarised in Table 1 with all other minor and moderate effect QTLs provided in Appendix A. Mapping studies conducted in recombinant inbred lines (RILs) derived from USG3209 × Jaypee found 48 QTLs clustering into 10 genomic regions in greenhouse and field trials. Three QTLs were identified on chromosome 1BL under waterlogging field and waterlogging greenhouse conditions, which explained 22–32% PV. Another major QTL (*QSpad3.ua-1D.5*) on chromosome 1D for chlorophyll content explained 24% PV under control greenhouse conditions [122]. In the population of W7984/Opata85, 36 QTLs associated with agronomic traits, including root/shoot dry weight index and total dry weight index, were identified across 18 chromosomes under waterlogged conditions, which explained up to 28.2% of the PV [123]. Thirty-two QTLs have been identified for waterlogging related traits, including survival rate, germination rate index, leaf chlorophyll content, plant height index, and dry matter weight in a W7984/Opata85 population. The major QTL on chromosome 7A was associated with germination rate index under waterlogging conditions, explaining 23.92% of PV [124].

### 5.4. Barley

In barley, several major effect QTLs were identified for survival rate, leaf chlorosis, root porosity, and aerenchyma formation under waterlogged conditions [52,104,110], which are summarised in Table 1 and Appendix A. Three QTLs (*KWw2.1*, *GSw1.1*, *GSw2.1*) associated with kernel weight and grains per spike on chromosome 2H explained 27.4–55.3% PV under waterlogged conditions in Yerong/Franklin population [125]. Other major QTLs (*tfy1.1-1*, *tfy1.2-1*, *tfy2.1-1*, *tfy1.1-2*) linked to leaf chlorosis were identified on chromosomes 2H, 3H and 4H in the TX9425/Franklin and Yerong/Franklin populations, explaining up to 36% PV [104,126]. For plant healthiness and survival rate under waterlogged conditions, the major QTLs (*QWI.YyFr.2H*, *QWL.YeFr.4H*, *QTL-WL-4H*) were detected on chromosomes 2H and 4H, which explained 23.9–30.1% PV [104,108,127]. The major QTL for root porosity and aerenchyma formation were identified in both populations YYXT/Franklin and Yerong/Franklin on chromosome 4H, explaining up to 44% PVE [110,111]. Another major QTL (GYw1.2) linked to grain yield was identified on chromosome 7H under waterlogged conditions in the population of Yerong/Franklin, explaining 30.4% PVE [125]. In addition, other moderate and minor effects QTLs are provided in Appendix A.

**Table 1 plants-10-01560-t001:** Summary of major QTLs associated with waterlogging tolerance in maize, rice, wheat, and barley.

Species	Chr	QTL	Trait	LOD ^a^	PVE (%) ^b^	Population ^c^	Population Size	Population Type	Reference
Maize	4	*-*	Leaf chlorosis	11.9–25.5	29.0–49.0	*Zea nicaraguensis* × Mi29	652	BC_3_F_4_	[113]
Maize	6	*Subtol6*	Mean leaf senescence score	-	22.0	Mo18W × B73	166	RILs	[114]
Maize	7	*Qarf7.04-7.05*	Adventitious root formation	5.1	21.0	B64 × Na4	110	F_2_	[115]
Maize	8	*Qarf8.05*	Adventitious root formation	7.0	25.0	B64 × *Zea mays* ssp. *huehuetenangensis*	186	F_2_	[56]
Maize	9	*sdw9-4*	Shoot dry weight	7.0	20.8	HZ32 × K12	288	F_2:3_	[43]
Maize	9	*tdw9-2*	Total dry weight	5.9	31.7	HZ32 × K12	288	F_2:3_	[43]
Maize	9	*tdw9-3*	Total dry weight	5.9	30.7	HZ32 × K12	288	F_2:3_	[43]
Rice	1	*qTIL1 ^C9285^*	Total internode elongation length	4.4	22.0	C9285 × T65	94	F_2_	[118]
Rice	1	*qTIL1 ^T65^*	Total internode elongation length	3.9	20.0	W0120 × T65	94	F_2_	[118]
Rice	9	*Sub1*	Green leaf recovery	36.0	69.0	IR40931-26 × PI543851	169	F_2_	[120]
Rice	12	*qTIL12 ^C9285^*	Total internode elongation length	6.2	27.0	C9285 × T65	94	F_2_	[118]
Rice	12	*qTIL12 ^W0120^*	Total internode elongation length	5.9	36.0	W0120 × T65	94	F_2_	[118]
Rice	12	*qNEI12 ^C9285^*	Number of elongated internodes	6.3	27.0	C9285 × T65	94	F_2_	[118]
Rice	12	*qNEI12 ^W0120^*	Number of elongated internodes	4.5	27.0	W0120 × T65	94	F_2_	[118]
Rice	12	*qLEI12 ^C9285^*	Lowest elongated internode	7.8	36.0	C9285 × T65	94	F_2_	[118]
Rice	12	*qLEI12 ^W0120^*	Lowest elongated internode	5.6	26.9	W0120 × T65	94	F_2_	[118]
Wheat	1BL	*QRfbio.ua-1B-WGH*	Root fresh biomass	6.6	22.0	USG3209 × Jaypee	130	RILs	[122]
Wheat	1BL	*QSfbio.ua-1B-WGH*	Shoot fresh biomass	6.7	27.0	USG3209 × Jaypee	130	RILs	[122]
Wheat	1BL	*QSpadpost.ua-1B-WF*	Chlorophyll content	4.8	32.0	USG3209 × Jaypee	130	RILs	[122]
Wheat	1D	*QSpad.ua-1D.5*	Chlorophyll content	3.0	24.0	USG3209 × Jaypee	130	RILs	[122]
Wheat	2B	*–*	Root/shoot dry weight	3.0–8.3	9.5–23.3	W7984 × Opata 85	112	RILs	[124]
Wheat	7A	*GRI-7A*	Germination rate index	2.9–7.6	11.4–23.9	W7984 × Opata 85	112	RILs	[124]
Barley	2H	*KWw2.1*	Kernel weight	9.1	27.4	Yerong × Franklin	156	DH lines	[125]
Barley	2H	*GSw1.1* *GSw2.1*	Grains per spike	11.5 5.6	35.4 55.3	Yerong × Franklin	156	DH lines	[125]
Barley	2H	*tfy1.1-1*	Leaf chlorosis	9.2	23.3	TX9425 × Franklin	92	DH lines	[126]
Barley	2H	*QWI.YyFr.2H*	Plant healthiness	18.7	30.1	YYXT × Franklin	172	DH lines	[127]
Barley	3H	*tfy1.2-1* *tfy2.1-1*	Leaf chlorosis	7.3 9.3	36.0 34.1	TX9425 × Franklin	92	DH lines	[126]
Barley	3H	*tfy1.1-2*	Leaf chlorosis	7.3	36.0	TX9425 × Franklin	92	DH lines	[126]
Barley	4H	*-*	Yellow leaf percentage	3.8–16.5	6.7–26.7	Yerong × Franklin	177	DH lines	[104]
Barley	4H	*QWL.YeFr.4H*	Survival rate	14.5	23.9	Yerong × Franklin	177	DH lines	[104]
Barley	4H	QTL-AER	Aerenchyma formation	51.4	76.8	TAM407227 × Franklin	163	DH lines	[108]
Barley	4H	*QTL-WL-4H*	Waterlogging tolerance	19.2	34.6	TAM407227 × Franklin	163	DH lines	[108]
Barley	4H	*–*	Root porosity Aerenchyma formation	6.4 9.4	25.6 44.0	Yerong × Franklin	177	DH lines	[111]
Barley	4H	*yfy2.2-3*	Leaf yellowing	10.4	22.4	Yerong × Franklin	177	DH lines	[126]
Barley	4H	*-*	Root porosity	12.1–13.5	35.7–39.0	YYXT × Franklin	126	DH lines	[110]
Barley	7H	GYw1.2	Grain yield	7.5	30.4	Yerong × Franklin	156	DH lines	[125]

^a^ LOD, logarithm of the odds; ^b^ proportion of phenotypic variance explained; ^c^ mapping population, boldface indicates the parent contributing the favourable allele.

### 5.5. Challenges on QTL Mapping

Most QTL mapping studies were conducted in different types of mapping populations derived by crossing two parents with a contrasting phenotype (waterlogging tolerant vs. susceptible). The main limitations of such biparental populations include (a) the presence of relatively few recombination events that often allow the localisation of QTL with a large confidence interval, and (2) the ability to detect QTL depends on the phenotypic diversity of the two parents, which may constitute a small part of the genetic variation in the species. To minimise limitations associated with biparental populations, multi-parent advanced generation intercross (MAGIC) strategy has been applied to provide higher recombination and mapping resolution by multiple alleles introgression [128]. In rice, a MAGIC population was produced by intercrossing of eight *indica* parents and used to detect QTLs for waterlogging tolerance [129]. In maize, a MAGIC population derived from eight genetically diverse lines was used to map QTLs and compared with the nested association mapping (NAM) population. The MAGIC population showed a higher mapping power than the NAM population that shared a common parentage [130]. In barley, a MAGIC population was derived from eight spring genotypes, which was used to map and characterise a flowering-time gene *Vrn-H3* from QFT.MAGIC.HA-7H.A [131]. Assessment of diverse phenotypic traits including disease resistance and yield potential in the wheat MAGIC population found all traits exhibited larger genetic diversity [132]. Although MAGIC population provided greater genetic diversity and better resolution for QTL mapping, most QTLs were found to be genetic background specific or highly influenced by the environment and G x E interactions. As we described above, many QTLs were detected under a specific condition, suggesting the need for evaluation across multiple environmental conditions to better estimate their effect and stability before consideration for their use in breeding programs.

## 6. Candidate Genes for Waterlogging Tolerance

### 6.1. Maize

The gene expression analysis on maize identified differentially expressed genes under waterlogging stress involved in the generation or scavenging ROS, Ca^2+^ signalling, and cell wall loosen degradation pathways. *GRMZM2G300965* (*RBOH*) is involved in ROS generation and showed 117-fold higher expression levels in cortical and stelar cells under waterlogging stress compared to controlled conditions. *GRMZM2G174855* (*XET*) also exhibited increased expression in cortical and stelar cells under waterlogged conditions [96]. Two genes were identified in the region of chromosome 5: *GRMZM2G463640* (cytochrome b6 gene) that is oxygen dependent and changes its structure under oxygen deficiency [133], and *GRMZM2G095239* (a single myb histone 6 gene) which is involved in inducing *ADH* expression under insufficient oxygen conditions [134]. *GRMZM2G053503* (AP2 domain-containing protein) is similar to Etheylene Responsive Transcription Factor (ERF) and was expressed 30-fold higher under waterlogged conditions [96]. In addition, *GRMZM2G055704* (heavy metal transport/detoxification superfamily protein, HMS), located on chromosome 1, was identified as a waterlogging tolerant candidate gene using BSR-seq to differentiate gene expression in 10 susceptible and 8 tolerant inbred lines of maize [5]. Two candidate genes are underlying the QTL *Subtol6*, *RAV1* and *HEMOGLOBIN2 (HB2)*, and are known to regulate leaf senescence in *Arabidopsis* and suppress ROS levels in maize, respectively [114]. The gene *ZmERB180*, belonging to the group VII ethylene response factors contributes to waterlogging tolerance. Ectopic expression of *ZmERB180* in *Arabidopsis* and overexpressed transgenic lines in maize increased survival rate under waterlogging stress via adventitious roots formation and ROS homeostasis [135]. *Vitreoscilla* haemoglobin (VHb) is a type of haemoglobin in the aerobic bacterium *Vitreoscilla*, which has been found to contribute to anaerobic stress tolerance in plants. The transgenic line with the introduction of VHb of *Arabidopsis* and maize exhibited significantly improved growth traits (seedling length, primary root length, lateral root number, shoot dry weight) compared with control [136].

### 6.2. Rice

Genetic and molecular mechanisms of waterlogging tolerance have been extensively studied in rice. Two opposite strategies quiescence and escape of rice are controlled by different genes, respectively. *Sub1* is located on chromosome 9 and contains a cluster of three ERF genes: *Sub1A*, *Sub1B*, and *Sub1C*. *Sub1A* suppresses internode elongation and induces fermentative metabolism under waterlogging stress. The submergence tolerant variety FR13A containing *Sub1A* does not elongate under waterlogging conditions but regrowth after water recession, while other varieties that lack *Sub1A* rapidly grow under submergence to escape the stress [137]. *Sub1A* controls quiescence strategy by stabilising GA signalling repressors Slender rice-1 (SLR1) and SLR1-Like-1 (SLRL1) proteins to inhibit elongation under waterlogging stress [138]. Introducing *Sub1A* from waterlogging-tolerant lines to intolerant lines has improved submergence tolerance through increased expression of the *ADH1* gene, suggesting that *Sub1A* is the most effective gene for waterlogging tolerance [121]. Moreover, three ERF genes in *Arabidopsis*, *At1g72360* (*HRE1*), *At2g475220* (*HRE2), At3g14230 (RAP2.2)*, belong to the same ERF group (ERFVII) as *Sub1A*. The function of these three genes under hypoxic conditions was confirmed by gene overexpression and knockout [139,140,141], which can further validate *Sub1A* role in submergence tolerance.

In contrast, the induction of *Sub1A* expression inhibits escape strategy controlling by ethylene-induced GA elongation. The escape strategy is controlled by *SNORKEL1 (SK1)* and *SNORKEL2 (SK2)* which positively regulate GA synthesis to control internode elongation. The *SK* genes are strongly induced by ethylene accumulation, leading to internodes elongation mediated by GA signalling during waterlogging [142]. The transgenic plants with overexpressed *SK1* and *SK2* genes elongated one to three and seven internodes, respectively [142].

### 6.3. Wheat

Several candidate genes have been reported in wheat, which were involved in ROS-producing/scavenging to regulate aerenchyma formation in roots. ROS accumulation is common under waterlogging stress; antioxidant enzymes such as catalase (CAT) and superoxide dismutase (SOD) inhibited ROS accumulation to cope with the stress [143]. SOD can be classified into *Cu/Zn-SODs*, *Fe-SODs*, and *Mn-SODs* according to the associated metal ion, which is the first enzyme for ROS scavenging as it catalyses O_2_^-^ into H_2_O_2_ followed by H_2_O_2_ degradation catalysing by CAT [144]. For wheat variety Huamai 8, the gene expression level of *NADPH oxidase* and *SOD* encoding ROS-producing enzymes were significantly increased after 12 h waterlogging treatment and then decreased at 24 and 48 h. Their counterparts, catalase (*CAT*) and metallothionein (*MT*) encoding ROS-scavenging enzymes showed strongly inhibited expression levels after 12 h waterlogging treatment, and then the reduced repression was observed at 24 and 48 h [51]. However, a waterlogging study conducted in the wheat cultivars Hua 8 (tolerant) and Hua 9 (susceptible) showed differential gene expression patterns under waterlogged conditions. In Hua 9, waterlogging stress inhibited *CAT* expression and activity, reduced *MnSOD* expression, and SOD activity. However, in Hua 8, waterlogging treatment increased *CAT* expression except 12 days after flowering and induced *MnSOD* expression, SOD activity. Longer waterlogging treatment caused *SOD* and CAT activity to strongly decrease in Hua 9 and increase in Hua 8, suggesting Hua 8 have a stronger capability for ROS homeostasis [145].

Group VII of ERF (ERFVII) contains conserved N-terminal domain (MCGGAI/L) of targeted proteolysis, which is an important mechanism for hypoxia response in plants. ERFVII are substrates of the N-end rule pathway and sense oxygen by oxidation of tertiary destabilising cysteine (Cys) residue [146,147,148]. As oxidation of Cys, arginylation, and ubiquitination under aerobic conditions result in ERFVII degradation, hypoxia can stabilise these proteins and increase survival rate under waterlogging stresses [146,149]. *TaERFVII.1* showed different gene expression patterns between waterlogging tolerant and susceptible wheat varieties. *TaERFVII.1* silencing lines influenced the expression of waterlogging-responsive genes, while constitutive and stabilised expression *TaERFVII.1* with MYC-peptide tag at its N terminus improved waterlogging tolerance by increased survival rate, chlorophyll content of leaf, and induced waterlogging-related genes in wheat [150]. In addition, among 23 pathogenesis-related (PR) protein-1-like genes, TaPR-1.2 is involved in the stress (humidity)-responsive process, and it was significantly increased during lysigenous aerenchyma formation [151]. Overexpressed two TaPR-1.2 cDNA affected interested protein (ferredoxin, ribosomal translocation enzyme) levels under waterlogging stress [152].

### 6.4. Barley

Several candidate genes have been identified in barley associating with aerenchyma formation and energy metabolism. Waterlogging stress significantly induced *XET*, *RBOHD*, *PDC* expression in roots with a more pronounced effect for tolerant varieties [18]. Ethylene synthesis genes (*ACC oxidase*), ROS corresponding genes (*RBOH*), and cell-wall-loosening enzyme (*XET*) regulate aerenchyma formation via cortical programmed cell death to cope with waterlogging stresses (as described in Section 3.3.1 Phytohormone signalling). Proteomic studies on barley have identified three candidate genes for waterlogging tolerance: *PDC*, *ACC oxidase*, and *GST* as they are induced by waterlogging stress in the roots of both barley susceptible and tolerant varieties, with higher expression in tolerant varieties [70]. *PDC*, *ACC oxidase*, and *GST* are involved in ATP production, ethylene synthesis, and ROS-scavenging pathway, respectively [74]. Of these three genes, *GST* and *ACC oxidase* are located in the identified QTL region for waterlogging responsive traits, which was mapped on chromosome 7H [108]. *GST* was mapped on chromosome 4H located within the major QTL for aerenchyma formation and waterlogging tolerance [110]. The genes co-localised with previously identified QTLs in barley may have a major function in controlling the waterlogging tolerance mechanism.

BERF1, a member of ERFVII family in barley, is a substrate of the N-end rule pathway in vivo, suggesting barley ERFVII regulate waterlogging response by oxygen sensor Nt-Cys. PROTEOLYSIS6 (PRT6) is N-recognin for arginine branch of the N-end rule pathway and catalyses proteosome degradation. Physiological analyses found that reduced HvPRT6 level contributed to waterlogging tolerance by stabilising growth and greater chlorophyll retention [148]. Genome-widely analysed ERF gene family in barley found *HvERF2.11* was highly induced by waterlogged conditions in waterlogging tolerant line TF58. Transgenic line of *Arabidopsis* with overexpressed *HvERF2.11* exhibited higher activities of antioxidant enzyme and ADH and more tolerant for waterlogging stress, which further confirmed the positive regulatory role of *HvERF2.11* [153].

## 7. Identification of Syntenic Candidate Genes for Waterlogging Tolerance in Crops

Only limited knowledge is available on the genetic mechanisms and functional characteristics of genes controlling waterlogging response in wheat, barley, maize. Previously identified functional waterlogging genes in rice and the model plant *Arabidopsis* have provided some clues for identifying the syntenic candidate genes for waterlogging tolerance in the genetically more complex maize, barley, and wheat. For example, barley root porosity QTL on chromosome 4H is syntenic with rice *Sub1A-1* on chromosome 9 and maize aerenchyma formation QTL (*Qaer1.02-3*) on chromosome 1 [110].

We summarised five genes from rice (*Sub1A*, *Sub1B*, *Sub1C*, *SNORKEL1*, *SNORKEL2*), six genes from Arabidopsis (*WRK40*, *WRK45*, *HRE1*, *HRE2*, *RAP2.2*, *RAP2.12*), and five genes from maize (*GRMZM2G055704*, *ZmEREB180*, *GRMZM2G416632*, *GRMZM2G300965*, *GRMZM2G053503*) associating with waterlogging tolerance to identified 65 syntenic genes in wheat, barley, and maize. These genes are involved in WRKY transcription factor, glutathione S transferase, respiratory burst oxidase homolog, heavy metal transport/detoxification superfamily protein, and ethylene-responsive factors. The detailed location of these genes is shown in Figure 2 and the detailed gene information is shown in Appendix A.

## 8. Breeding for Waterlogging Tolerance

Field screening for waterlogging tolerant lines has been carried out in wheat and barley [154]. In addition, marker-assisted selection for *Sub1* allele has been used in rice breeding programs [155]. Using RM484, RM23887, SSR1, and 18 flanking markers for screening recombinant lines, the recipient variety Swarna has been converted to a submergence tolerant variety after three backcross generations [156]. However, the use of QTL-associated molecular markers for complex quantitative traits was found to be more challenging for multiple reasons described in the previous sections, which includes a small phenotypic effect of most QTLs, populations and environment specificity of most QTLs, limited genetic diversity in most biparental populations, low correlation between recombination frequency-based genetic maps and physical maps, and epistasis [157]. Genomic selection has been frequently cited as an alternative method for developing waterlogging tolerant varieties. Genomic selection was applied to predict genetic value by combining genome-wide marker and phenotyping data of a training population to obtain genomic estimated breeding value (GEBV) [158]. In maize, rapid-cycle genomic selection (RCGS) for waterlogging tolerance in two multi-parent yellow synthetic populations showed a genetic gain of 38–113 kg ha^−1^ year^−1^ under waterlogging stress [159]. The results suggested that RCGS is an effective approach to breed varieties with superior qualities by introgressing parents with traits of interest [160]. However, the prediction accuracies of genomic selection depend on multiple factors, including models, population size, trait heritability, the relationship between training and prediction sets [161]. Gene editing technologies have opened a new avenue for breeding programs. The integration of CRISPR (clustered regularly interspaced short palindromic repeats)/Cas (CRISPR associated proteins)-based gene editing can be used to modify targeted sequences, such as loss-of-function, gain-of-function, and altered expression [162]. Genomic editing using CRISPR/Cas system has been successfully applied in rice, wheat, maize, and barley to improve grain yield, quality, biotic, and abiotic stresses as well as herbicide resistance [163,164]. The CRISPR/Cas system can be used to develop improved varieties once waterlogging functional genes are characterised.

## 9. Conclusions and Future Prospect

The identification of waterlogging tolerant genotypes and understanding the traits, genes, and QTLs associated with waterlogging tolerance are critical in developing waterlogging tolerant rice, barley, maize, and wheat varieties. Some of the genes (e.g., *Sub1*) and major effect QTL (those with R^2^ > 20%) that we highlighted in our review are good candidates for developing new varieties using marker-assisted selection. However, most QTLs have been reported to be population and environment-specific, which restricts their use in breeding programs. For such reasons, new technologies, including genomic selection, gene editing, and omic sciences would play an important role in developing waterlogging tolerant varieties. In maize, wheat, and barley, most studies thus far are focusing on gene expressions under waterlogged conditions, but functional genes for waterlogging tolerance have not yet been identified. More studies are needed to (1) characterise the function of syntenic waterlogging tolerant genes from *Arabidopsis* and rice as well as differentially expressed genes under waterlogged conditions in maize, wheat, and barley; (2) validate and fine map some of the major effects QTLs identified in each crop; (3) develop breeder friendly molecular markers for some of the promising QTLs for use in marker-assisted breeding; (4) evaluate the predictive ability of genomic selection under waterlogged conditions; (5) conduct extensive germplasm characterisation in each of the cereals crops to identify new source of waterlogging tolerance.

## Figures and Tables

**Figure 1 plants-10-01560-f001:**
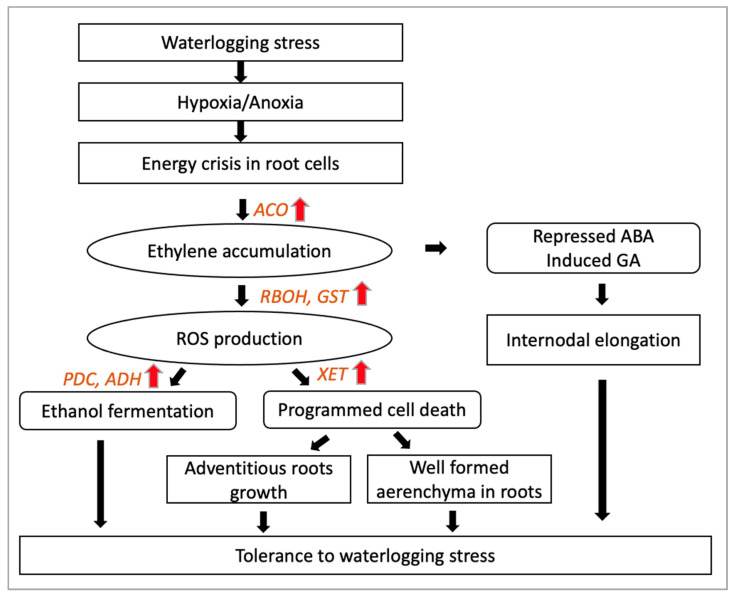
Schematic diagram of the main waterlogging stress responses and metabolic adaptive traits for waterlogging tolerance in plants. PDC: pyruvate decarboxylase, ADH: alcohol dehydrogenase, RBOH: respiratory burst oxidase homolog, GST: glutathione S transferase, XET: xyloglucan endo-transglycosylase, ACO: 1-amino-cyclopropane-1-carboxylic acid oxidase.

**Figure 2 plants-10-01560-f002:**
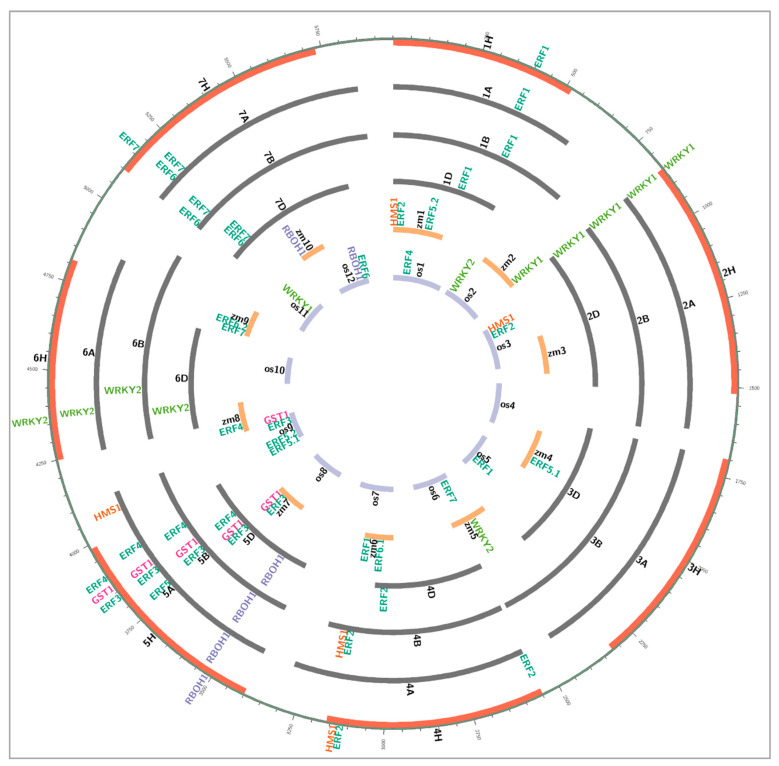
Homologous candidate genes for waterlogging tolerance in rice, maize, barley, and wheat. WRKY: WRKY transcription factor family, GST: glutathione S transferase, RBOH: respiratory burst oxidase homolog, HMS: heavy metal transport/detoxification superfamily protein, ERF: Ethylene-responsive factor. Each bar represents the individual chromosome, and the chromosome name is placed in the middle of the bar. The purple bars indicate 12 chromosomes of rice; the orange bars indicate 10 chromosomes of maize; the grey bars indicate 7 chromosomes of wheat A, B, D genome; the red bars indicate 7 chromosomes of barley. Gene name placed on the chromosome bar based on the physical position.

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
