# Peer review of "Opportunities for Improving Waterlogging Tolerance in Cereal Crops—Physiological Traits and Genetic Mechanisms"

_plants, 2021, doi:10.3390/plants10081560_

Round 1
Reviewer 1 Report
The manuscript summarized QTL results and candidate genes for waterlogging tolerance in major crops. Because waterlogging caused by climate change in worldwide is drastically affect on crop production, the review manuscript is very interesting for readers, Plants. To improve the manuscript, please reconsider following points.
P2L38: This sentence described barley, however, reference (4) is focused on maize. Also, page number in reference (4) should be “59-72”, not “39-72”. I found such mistakes in many parts of the manuscript. I would advise the authors to check the manuscript carefully.
P5L182: I think “spongy” should be deleted because we use this term for secondary aerenchyma.
P5L196-197: In this description correct? (i.e., positive correlation in barley)
P6L218: “Radial oxygen loss (ROL) barrier” instead of “Radial oxygen loss (ROL)”.
P11 Table 1: Names of the QTLs have been updated (e.g., Qarf3.07-3.08 instead of 3.07-8). The authors can find at https://doi.org/10.1270/jsbbs.20117
In addition, please check carefully. For example;
Qaer2.06: “Zea” instead of “Zea.” (delete “.”).
Qaer3.10: “Zea luxurians” instead of “B73”.
Qaer5.09: “B64” instead of “Zea nicaraguensis”.
Qaer8.06-7: “Overdominance gene action“ or “B64” instead of “Zea nicaraguensis”.
8.03: “huehue” instead of “Huehue”.
P13L418: The value of “0.8” seems to be strange. Such QTL cannot be detected.
P15L440-442: Please provide reference.
P15 Table 4: Franklin is a sensitive variety, however, this variety contributes positive effect in many QTLs. I understand this situation (due to scoring for trait evaluation) but readers may be confused.
P22: Table S1: Names of gene in 4 species are the same (e.g., GST1), but in maize, “GST23” instead of “GST1”.
Author Response
Reviewer 1
Comment 1:
P2L38: This sentence described barley, however, reference (4) is focused on maize. Also, page number in reference (4) should be “59-72”, not “39-72”. I found such mistakes in many parts of the manuscript. I would advise the authors to check the manuscript carefully.
Answer 1:
We have changed refence (4) in the new version of the manuscript, please find the corrected reference on P30L782.
We have also checked and corrected (if necessary) page numbers for all other references.
Comment 2:
P5L182: I think “spongy” should be deleted because we use this term for secondary aerenchyma.
Answer 2:
The word “spongy” was replaced by “airy” (P5L200).
Comment 3:
P5L196-197: In this description correct? (i.e., positive correlation in barley)
Answer 3:
To address this comment, the sentence was changed to “The aerenchyma formation and higher root porosity are important adaptive traits contributing to waterlogging tolerance” on P6L215-216.
Comment 4:
P6L258: “Radial oxygen loss (ROL) barrier” instead of “Radial oxygen loss (ROL)”.
Answer 4:
Corrected, please find on P7 L250.
Comment 5:
P11 Table 1: Names of the QTLs have been updated (e.g., Qarf3.07-3.08 instead of 3.07-8). The authors can find at https://doi.org/10.1270/jsbbs.20117
In addition, please check carefully. For example;
Qaer2.06: “Zea” instead of “Zea.” (delete “.”).
Qaer3.10: “Zea luxurians” instead of “B73”.
Qaer5.09: “B64” instead of “Zea nicaraguensis”.
Qaer8.06-7: “Overdominance gene action“ or “B64” instead of “Zea nicaraguensis”.
8.03: “huehue” instead of “Huehue”.
Answer 5:
We have updated the QTL name in Table 1, and have also addressed all other points.
Comment 6:
P13L418: The value of “0.8” seems to be strange. Such QTL cannot be detected.
Answer 6:
The referenced paper describes a wide range of phenotypic variation explained by 36 QTLs from 0.8 to 28.2%. To address this comment, we changed the sentence to “Under waterlogged conditions, thirty-six QTLs for agronomic traits including root/shoot dry weight index and total dry weight index were identified on 18 chromosomes in W7984/Opata85 explaining PVE of phenotypic variation of up to 28.2%” (P15 L499-P16L502).
Comment 7:
P15L440-442: Please provide reference.
Answer 7:
Reference 113 was added (P17 L530).
Comment 8:
P15 Table 4: Franklin is a sensitive variety, however, this variety contributes positive effect in many QTLs. I understand this situation (due to scoring for trait evaluation) but readers may be confused.
Answer 8:
Actually, the detected positive effects of Franklin are most responsible for yellowing leaves and biomass reduction, which means that DH lines with alleles inherited from Franklin exhibit more waterlogging susceptible traits.
Comment 9:
P22: Table S1: Names of gene in 4 species are the same (e.g., GST1), but in maize, “GST23” instead of “GST1”.
Answer 9:
Table S1 provides the detailed information of Figure 2. GST1 in maize presents the homologous gene of GRMZM2G416632 (GST gene) followed by the gene name (Zm0001d015515_P001). GST1 of table S1 presents homologous gene of GST in four species rather than the specific annotation of different species, and corresponds with Figure 2.
Reviewer 2 Report
- Coordinate words are need to add "and", like in the line 60-61: Phytohormones signalling (ethylene, abscisic acid, gibberellin), line 65-66: cereal crops rice (Oryza sativa L.), maize (Zea mays L.), wheat (Triticum aestivum L.), barley (Hordeum vulgare L.)., etc.
- The abbreviation of "ribulose bisphosphate carboxylase" needs to check
- line 194-195:"ROS as well as enzymes involving in antioxidant defence systems are induced by waterlogging to deal with the damaging effects of oxidative stress" is confusing.
- line 320-322:"To avoid ROS damage, ROS scavengers, glutathione S-transferase (GST) protect plants from oxidative stress which was found upregulated in barley roots under waterlogging stress" is confusing.
Author Response
Reviewer 2
Comment 1:
Coordinate words are needed to add "and", like in the line 60-61: Phytohormones signalling (ethylene, abscisic acid, gibberellin), line 65-66: cereal crops rice (Oryza sativa L.), maize (Zea mays L.), wheat (Triticum aestivum L.), barley (Hordeum vulgare L.)., etc.
Answer 1:
We than the reviewer for pointing this out. We have now added coordinate words throughout the manuscript. Please find corrected sentences on P2 L60 and L65.
Comment 2:
The abbreviation of "ribulose bisphosphate carboxylase" needs to check
Answer 2:
The abbreviation of “ribulose bisphosphate carboxylase” was changed to “RuBisCo” in several instances, including on P4 L127 and P4L129.
Comment 3:
line 194-195:"ROS as well as enzymes involving in antioxidant defence systems are induced by waterlogging to deal with the damaging effects of oxidative stress" is confusing.
Answer 3:
To address this comment, this sentence was changed to “To cope with the adverse effects of ROS accumulation, antioxidant defence systems are employed in response to waterlogging stress” on P6 L220-221
Comment 4:
line 320-322:"To avoid ROS damage, ROS scavengers, glutathione S-transferase (GST) protect plants from oxidative stress which was found upregulated in barley roots under waterlogging stress" is confusing.
Answer 4:
To address this comment, the sentence was changed to “Glutathione S-transferase (GST) is a detoxification enzyme catalysing glutathione-dependent detoxification reaction, which was induced in waterlogged barley roots to minimize the damage of ROS accumulation” on P10 L384-L386
Reviewer 3 Report
This work describes the detrimental effects of waterlogging on physiological and genetic mechanisms in the model plant Arabidopsis and the major cereal crops rice, maize, wheat, and barley. In addition, describe conventional options and a few genetic engineering examples to develop waterlogging-tolerant varieties. Thirdly, it describes identified candidate genes for controlling waterlogging tolerance in Arabidopsis and rice to identify homologous genes in the less waterlogging-tolerant crops maize, wheat, and barley.
However, as this is a review article author must provide detail and comprehensive reports or studies for claim Arabidopsis and cereal crops rice, maize, wheat, and barley. It's not just a survey of information. The author mentioned in the abstract that they "discuss conventional and genetic engineering as options to develop waterlogging-tolerant varieties once target genomic regions are known" but they discussed few examples/studies on genetic engineering on Arabidopsis and lack the proper discussion on other crops rice, maize, wheat, and barley. Similarly, some of the sections represented studies from Arabidopsis, wheat, barley, maize but lack reports on rice, and some section has given limited information. Hence, this article needs substantial revision before it goes for publication.
Apart from the above comments, I have some observations and suggestions below
Introduction
As Author mentioned botanical or Scientific names for Arabidopsis, Barly, Rice, and Maize it's better to mention at first appearance than follow with the common name. Rice has appeared in the second paragraph so the author should mention it with a Scientific name followed by common names this needs to follow throughout the MS.
Line 64, Page 2, after Arabidopsis thaliana, give common name as Arabidopsis and then use the common name as suggested above.
Line 65, Page 2, (Oryza sativa L.), maize (Zea mays L.), wheat (Triticum aestivum L.), barley (Hordeum vulgare L.) no need to write again scientific name just use the common name as scientific names already mentioned in the previous para.
Line 81, Page 2, "Glucose is the main product of photosynthesis, which combined oxygen to pro- duces energy for plant growth by respiration" this sentence is not correct or clear. Glucose and oxygen are the basic or main products of photosynthesis hence please revise the sentence.
2.1 Oxygen deprivation under this section inhibition of root growth under waterlogging conditions discussed with few studies from barley, wheat, and maize but it lacs the studies from Arabidopsis and rice as the author mentioned this review is based on Arabidopsis and the major cereal crops rice, maize, wheat. So the author not discuss any study from Arabidopsis and rice please explain? Similarly, some other sections also lack examples or study reports for Rice please include them.
Line 107-108, Page 3, "Apart from stomatal closure, the decreased photosynthesis rate is also caused by mesophyll cell ultrastructure and chlorophyll content". This sentence is incomplete and not clear please revise it.
Line 126 Page 4, Root hydraulic conductance, appears in first sentence hence (Lp) term should be used at first appearance. Please revise it.
Line 153, Page 4, the author mentioned nitrogen (N), magnesium (Mg), copper (Cu), manganese (Mn) but not written full name for P, K, Zn, maintain the uniformity while writing.
Section "Nutrient absorption" lacks the studies reports from Arabidopsis and cereal crops rice, and wheat.
Under the Adaption or Anatomical adaption author discusses some traits, Is there any role or reports on deep rooting and root angle in waterlogging stress?
Please provide the full form of NAD
Line 294, Page 8, "GA and ABA act as antagonists in plant growth responses. It was found that reduced ABA synthesis and GA signalling by waterlogging stress induced shoot elongation in rice" This sentence is contradictory please revise the second sentence to make it clear.
Waterlogging stress is a complex quantitative trait that is regulated by multiple genes, and is also greatly influenced by environmental conditions, thus the author not mentioned anything about environmental effects or G x E studies please explain.
Commonly people use phenotypic variation as (PVE) it's better to use this as it was used several times.
Table 1 under "Population size" some studies just mentioned lines whereas some places generation like F2, BCF2, or F2:3 mentioned please explain why?
Line 511, Page 18, just use ROS no need to give full name as you have mentioned previously.
Page 20, Figure 2 legends need to explain with more detailed information like each bar and color represent for what?
Line 589 page 21, diagnostic markers term usually suits in medical section please use the alternative word for "diagnostic"
Conclusion: Author mentioned marker-assisted selection but not provided any reports or information on given crops in which markers are associated with traits and used for waterlogging stress tolerance screeing. In addition, not included any suggestions as a breeding perspective for environmental effect studies that need to conduct or some other suggestions for breeders.

Author Response
Comment1.1: 1:
However, as this is a review article author must provide detail and comprehensive reports or studies for claim Arabidopsis and cereal crops rice, maize, wheat, and barley. It's not just a survey of information.
Answer 1.1:
The purpose of a literature review is to gain an understanding of the existing research relevant to a particular field, and to present that knowledge in the form of a written report to help build current knowledge in the field. In this review we aim to provide an overview on the detrimental effects of waterlogging on physiological and genetic mechanisms in the model plant Arabidopsis and the major cereal crops rice, maize, wheat, and barley. We believe we have achieved our aims in this review, and without more information on the exact issue the reviewer has we do not know how to make further improvements.
Comment 1.2:
The author mentioned in the abstract that they "discuss conventional and genetic engineering as options to develop waterlogging-tolerant varieties once target genomic regions are known" but they discussed few examples/studies on genetic engineering on Arabidopsis and lack the proper discussion on other crops rice, maize, wheat, and barley.
Answer 1.2:
In the cases of maize, wheat, and barley, only relatively few studies are available on the topic of confirmed gene function for waterlogging using genetic engineering which were added in section 6 on P19-P23. We believe we have found all relevant available literature published to date on that topic and presented it in sufficient depth in our review in Sections 5 and 6. In addition to the written sections, we summarised all available QTL information (not all underlying candidate genes have been confirmed yet, and thus we can only report on QTL) in Tables 1-4, each table presenting detailed information on identified loci in each of the four target plant species we focussed on. We made sure that each of the four subsections in Section 6 are of equal length for each of the four plant species. In addition to that, we also produced a circos plot (Figure 2) visualising all detected candidate genes for all four species. We believe we not only summarised all available genetic information, but further have provided a great and up-to-date resource for researchers interested in waterlogging tolerance in cereal crops.
Comment 1.3:
Similarly, some of the sections represented studies from Arabidopsis, wheat, barley, maize but lack reports on rice, and some section has given limited information. Hence, this article needs substantial revision before it goes for publication.
Answer 1.3:
To address this comment, we added more information on Arabidopsis and cereal crops in section 2 physiological response of waterlogging stresses and section 3 adaption to waterlogging stresses on P2-P7.
Comment 2:
As Author mentioned botanical or Scientific names for Arabidopsis, Barley, Rice, and Maize it's better to mention at first appearance than follow with the common name. Rice has appeared in the second paragraph so the author should mention it with a Scientific name followed by common names this needs to follow throughout the MS.
Answer 2:
We have corrected this on P2.
Comment 3:
Line 64, Page 2, after Arabidopsis thaliana, give common name as Arabidopsis and then use the common name as suggested above.
Answer 3:
We have corrected this on P2L63
Comment 4:
Line 65, Page 2, (Oryza sativa L.), maize (Zea mays L.), wheat (Triticum aestivum L.), barley (Hordeum vulgare L.) no need to write again scientific name just use the common name as scientific names already mentioned in the previous part.
Answer 4:
Please fine corrections on P2L64.
Comment 5:
Line 81, Page 2, "Glucose is the main product of photosynthesis, which combined oxygen to pro- duces energy for plant growth by respiration" this sentence is not correct or clear. Glucose and oxygen are the basic or main products of photosynthesis hence please revise the sentence.
Answer 5:
To address this comment, the sentence was changed to “Glucose is the primary fuel for glycolysis and downstream pathways such as respiration to produce energy for plant growth and reproduction” on P2L80-81.
Comment 6:
“2.1 Oxygen deprivation” - under this section inhibition of root growth under waterlogging conditions discussed with few studies from barley, wheat, and maize but it lacks the studies from Arabidopsis and rice as the author mentioned this review is based on Arabidopsis and the major cereal crops rice, maize, wheat. So the author not discuss any study from Arabidopsis and rice please explain? Similarly, some other sections also lack examples or study reports for Rice please include them.
Answer 6:To address this comment, we added “Hypoxia also decreased root elongation and dry weight in both lowland and upland rice varieties 20” on P3 L98-99, which add more information on rice. “For Arabidopsis, lateral roots growth was inhibited and primary root growth direction changed under hypoxic conditions to explore more gas space in the soil 21,22” was on P3L99-101 to provide more information on Arabidopsis.
Comment 7:
Line 107-108, Page 3, "Apart from stomatal closure, the decreased photosynthesis rate is also caused by mesophyll cell ultrastructure and chlorophyll content". This sentence is incomplete and not clear please revise it.
Answer 7:
To address this comment, the sentence was changed to “Apart from stomatal closure, the photosynthetic rate is also decreased due to damaged mesophyll cell ultrastructure and reduced chlorophyll content.” on P3L115-116
Comment 8:
Line 126 Page 4, Root hydraulic conductance, appears in first sentence hence (Lp) term should be used at first appearance. Please revise it.
Answer 8:
Please find the corrected sentence on P4L134
Comment 9:
Line 153, Page 4, the author mentioned nitrogen (N), magnesium (Mg), copper (Cu), manganese (Mn) but not written full name for P, K, Zn, maintain the uniformity while writing.
Answer 9:
Please find the corrected sentence on P4L163-164.
Comment 10:
Section "Nutrient absorption" lacks the studies reports from Arabidopsis and cereal crops rice, and wheat.
Answer 10:
To address this comment, we added the following “The nutrient uptake by wheat seminal roots was lower in stagnant solution compared with aerated conditions 38. In barely root mature zone, the hypoxia immediately decreased net K+ uptake within few minutes 39.Consistently, mutant lines of Arabidopsis lacking K+ efflux channel were more capable of retaining K+ and showed higher tolerance to hypoxia 40. Waterlogging stress also decreased N metabolism and assimilation at different growth stage in maize 41. For rice, it copes with waterlogging stress by forming physical barrier to avoid oxygen diffusion from roots, which impairs nutrient absorption by roots 42.” on P4 L165-P5L172.
Comment 11:
Under the Adaption or Anatomical adaption author discusses some traits, Is there any role or reports on deep rooting and root angle in waterlogging stress?
Answer 11:
To address this question, we added the following: “Hypoxia changed the primary root growth angle of Arabidopsis to avoid deeper soil layers 22. A recently study of rice found the direction of adventitious root growth is determined by hormone auxin gradient in root tips 60. Adventitious roots grow upward to get closer to oxygen-rich water surface to facilitate water and nutrient uptake from the upper surface of the waterlogged soil61,62.” (P6 L236-241).
Comment 12:
Please provide the full form of NAD
Answer 12:
This is now corrected on P7L288
Comment 13:
Line 294, Page 8, "GA and ABA act as antagonists in plant growth responses. It was found that reduced ABA synthesis and GA signalling by waterlogging stress induced shoot elongation in rice" This sentence is contradictory please revise the second sentence to make it clear.
Answer 13:
To address this comment, the sentence was changed to “GA and ABA act as antagonists in plant growth responses, and GA plays an important role in stimulating shoot growth while ABA inhibits root elongation in rice 86. The application of ABA in rice decreased the extent of internode elongation induced by ethylene and GA 95. For deep-water rice, ABA declined by 75% and GA1 increased four-fold within three hours of the waterlogged plants with ethylene treatment 96. The adventitious roots of waterlogged wheat showed reduction of gene expression level in ABA biosynthesis and stem node ABA content 64. The reduction of ABA content was found in leaves and roots of both tolerant and intolerant barley varietiesafter three weeks waterlogging treatment with greater reduction in tolerant varieties 18. ” on P9L341-L354
Comment 14:
Waterlogging stress is a complex quantitative trait that is regulated by multiple genes, and is also greatly influenced by environmental conditions, thus the author not mentioned anything about environmental effects or G x E studies please explain.
Answer 14:
To address this comment, we added the following “A study of Arabidopsis showed lower temperatures caused less damage under waterlogged conditions by regulation of hypoxia-related genes107. In wheat, the severity of damage under waterlogging stress was influenced by the depth of waterlogging. The tillering was decreased by 62%, 45%, and 24% when the water level at 0 cm,10cm, and 20 cm below soil surface 108. Phenotyping of barley waterlogging tolerant lines found environment showed greater effect on waterlogging score of double haploid (DH) lines at the early stages 109. Dynamic QTL analysis for waterlogging tolerance in maize found different treatment time influenced expression of QTL during adapted traits development 110. A genotype by environment (G × E) interaction study was conducted in lowland rice using 37 genotypes across 36 environments from 1994 to 1997 showing board adaption cross different environmental conditions 111. As plant phenotypes are sensitive to the environmental conditions, the tolerant varieties should be evaluated in multiple conditions to identify stable waterlogging adaptive traits112. ” (P11L404-L416).
Comment 15:
Commonly people use phenotypic variation as (PVE) it's better to use this as it was used several times.
Answer 15:
We thank the reviewer for pointing this out, we have now changed this throughout the manuscript. Pleased find the corrected sentence on P12L458 in the new version of the manuscript.
Comment 16:
Table 1 under "Population size" some studies just mentioned lines whereas some places generation like F2, BCF2, or F2:3 mentioned please explain why?
Answer 16: This information is correct and taken directly from the referenced paper. F2:3 means F2-derived F3, a population derived from selfing of F2 individuals, and is a commonly used and accepted filial code. We also added population type as a new column in Table 1-4.
Comment 17:
Line 511, Page 18, just use ROS no need to give full name as you have mentioned previously.
Answer 17:
Corrected, please find on P21L631
Comment 18:
Page 20, Figure 2 legends need to explain with more detailed information like each bar and colour represent for what?
Answer 18:
To address this comment, we added following “Each bar represents the individual chromosome, and the chromosome name is placed in the middle of bar. The purple bars indicate 12 chromosomes of rice; the orange bars indicate 10 chromosomes of maize; the grey bars indicate 7 chromosomes of wheat A, B, D genome; the red bars indicate 7 chromosomes of barley. Gene name placed on the chromosome bar based on the physical position.” on P24L712-716.
Comment 19:
Line 589 page 21, diagnostic markers term usually suits in medical section please use the alternative word for "diagnostic"
Answer 19:
"Diagnostic markers" was changed to “molecular markers” on P25L740.
Comment 20:
Author mentioned marker-assisted selection but not provided any reports or information on given crops in which markers are associated with traits and used for waterlogging stress tolerance screening. In addition, not included any suggestions as a breeding perspective for environmental effect studies that need to conduct or some other suggestions for breeders.
Answer 20:
To address this comment, we added more information on the rice sub1 gene “Sub1A was first identified in submergence-tolerance lowland rice form India and have been used in breeding program 157. Sub1 were applied marker assisted backcrossing to develop submergence tolerant rice cultivars. The marker RM484, RM23887, SSR1 and 18 flanking markers were used for recombinant line screening within 6.5 Mb. The recipient variety Swarna of Sub1 was efficiently converted to submergence tolerant variety after three backcross generation 158.” on P21L614-619).
For the suggestions of breeding perspective, we added “The consistent expression of QTL across environment is required for MAS to identify reliable markers 163. As plant genotypes of waterlogging tolerance interact with environmental conditions, the screening of waterlogging tolerant breeding lines should be conducted in a glasshouse under controlled conditions or carried out in fields for many years at different locations.” on P25L726-730.
Round 2
Reviewer 3 Report
I am satisfied with the corrections made by the Author towards the Improvement of this article. I would like to endorse the current form MS for publication. Prior to final submission please correct the font size Page 22, L608-613.